# Covid-19 in outpatients—Is fever a useful indicator for SARS-CoV-2 infection?

**Anne Schneider**[1,2], **Holger Kirsten**[ORCID][3,4], **Franziska Lordick**[5], **Florian Lordick**[6], **Christoph Lübbert**[ORCID][1,2,7‡], **Amrei von Braun**[ORCID][1,2‡] *

1 Division of Infectious Diseases and Tropical Medicine, Department of Medicine II, Leipzig University Medical Center, Leipzig, Germany, 2 Interdisciplinary Center for Infectious Diseases, Leipzig University Medical Center, Leipzig, Germany, 3 Institute for Medical Informatics, Statistics and Epidemiology (IMISE), Medical Faculty, Leipzig University, Leipzig, Germany, 4 LIFE Research Center for Civilization Diseases, Leipzig University, Leipzig, Germany, 5 Medical Faculty, Leipzig University, Leipzig, Germany, 6 Department of Medicine II (Oncology, Gastroenterology, Hepatology, Pneumology and Infectious Diseases), Leipzig University Medical Center, Leipzig, Germany, 7 Department of Infectious Diseases/Tropical Medicine, Nephrology and Rheumatology, St. Georg Hospital, Leipzig, Germany

‡ These authors share responsibility.
* amrei.braun@medizin.uni-leipzig.de

**Data Availability Statement:** All relevant data are within the manuscript and its Supporting Information files.

**Funding:** The authors acknowledge support from Leipzig University for Open Access Publishing.

## Abstract

### Objective

Understanding mild to moderate symptoms of coronavirus disease 2019 (Covid-19) is important in order to identify active cases early and thus counteract transmission.

### Methods

In March 2020, Leipzig University Hospital established an outpatient clinic for patients potentially infected with severe acute respiratory syndrome coronavirus 2 (SARS-CoV-2). Confirmed cases with mild to moderate symptoms self-isolated at home and were followed-up by daily telephone calls for at least 14 days. Symptoms and course of illness of these patients are reported here.

### Results

From March 20 to April 17, 2020, 1460 individuals were tested for SARS-CoV-2 by naso- or oropharyngeal swab for real-time polymerase chain reaction (RT-PCR). Covid-19 was confirmed in 91 (6.2%) patients, of which 87 were included in the final analysis. Patients presented for testing after a mean of 5.9 days (IQR = 2.0–8.5). The median age was 37.0 years (IQR = 28.5–53), and 48 (55.2%) were female. Five (5.7%) patients required hospital admission during the course of illness. Most frequently reported symptoms were fatigue (n = 64, 74%), cough (n = 58, 67%), and hyposmia/hypogeusia (n = 44, 51%). In contrast to previous reports, fever occurred in less than a third of patients (n = 25, 29%). By day 14, more than half of the patients had recovered completely (n = 37/70, 52.9%).

**Competing interests:** The authors have declared that no competing interests exist.

## Conclusions

Fever seems to be less common in patients of relatively young age diagnosed with mild to moderate Covid-19. This suggests that body temperature alone may be an insufficient indicator of SARS-CoV-2 infection.

## Introduction

On January 27, 2020, the first infection with severe acute respiratory syndrome coronavirus 2 (SARS-CoV-2) was diagnosed in Germany [1]. The index patient was a young woman of Chinese origin without known comorbidities and, of note, with only mild symptoms at the time of transmission, leading to a cluster of at least four consecutive infections in Bavaria [2]. Similar reports of transmission from asymptomatic or mildly symptomatic patients followed, and today it has become clear that transmission of SARS-CoV-2 starts early after infection and often before infected persons become symptomatic [3, 4].

The main symptoms of patients with Covid-19 include dry or productive cough, fever, and rhinitis [5, 6]. Hyposmia and hypogeusia following infection are reported in up to 88% of cases worldwide [7–10]. However, most reports so far focused on hospitalized patients with moderate to severe courses of Covid-19, while published reports focusing on outpatients with mild symptoms are scarce. As detection followed by self-isolation of patients early after infection is known to sufficiently halt the spread of SARS-CoV-2, recognizing symptoms in mild to moderately affected patients is an essential aspect of successful containment [11].

In line with other centers across Europe, the Leipzig University Hospital established a specific outpatient clinic for patients potentially infected with SARS-CoV-2 at the beginning of March 2020. According to the German national health policy at the time, all persons with symptoms suggestive of Covid-19, as well as asymptomatic members of so-called system-relevant professions (e.g. healthcare, police, fire brigade, energy supply) with direct contact to confirmed cases were eligible for SARS-CoV-2 testing by naso- or oropharyngeal swabs for real-time polymerase chain reaction (RT-PCR). As the majority of general practitioners in the greater Leipzig region were struggling to provide these services at the time due to a lack of personal protective equipment, diagnostics and follow-up care of patients diagnosed with Covid-19 was ensured by our clinic. Confirmed cases of Covid-19 with mild to moderate symptoms self-isolated at home and were followed up by daily telephone calls for at least 14 days in order to assess the course of illness and to react in case of progression requiring hospital admission. Symptoms and course of illness of these patients are reported here.

## Methods

This study was conducted at the designated Covid-19 outpatient clinic of the Leipzig University Hospital, Germany. Patients of all age groups diagnosed with Covid-19 by naso- or oropharyngeal swab for RT-PCR of SARS-CoV-2 (RealTime SARS-CoV-2 assay, Abbott Laboratories, Abbott Park, Illinois, USA) between March 20 and April 17, 2020, were eligible for inclusion. Swabs were done by trained medical staff and analyzed on the same day at the Institute of Virology of the Leipzig University Hospital. Symptoms and course of illness were assessed upon presentation, followed by daily telephone calls for at least 14 days using a standardized questionnaire. Patients requiring further evaluation were referred to the emergency care unit or directly to a dedicated Covid-19 inpatient department. Patients admitted for inpatient care on the day of diagnosis were not included in this study.

Data was accessed from April 30, 2020, onwards, and was derived from electronic patient files, patient questionnaires, and virology reports, and analyzed using the software SPSS Version 24.0 (IBM Corporation, Armonk, NY, USA) and R 3.6.2 [12]. Besides baseline patient characteristics, type and sequence of reported symptoms from the time of onset were described. We tested for a trend of different percentages of test-positive persons across age groups by applying the Cochran-Armitage test for trend, considering p-values of $\leq 0.05$ statistically significant. Furthermore, chronological appearance and duration of symptoms were analyzed, separating the patients in early vs. late presenters, defined as tested positive for Covid-19 $\leq 7$ days or $>7$ days after the onset of symptoms. Using the anamnestic date of symptom onset in all symptomatic patients instead of the day of diagnosis to investigate the disease course, the number of patients with available data varies for each day Differences of weekly presence of symptoms between groups were calculated using Fisher's exact test. Correction for multiple testing was done by calculating the false discovery rate (FDR) according to Benjamini & Hochberg accounting for the number of tested symptoms and tested weeks, considering a FDR$\leq 0.05$ as statistically significant [13]. Additionally, we assessed positive or negative correlations between symptoms occurring in the same individual by calculating the repeated measures correlation coefficient r considering r$\geq 0.3$ as moderately strong correlation [14]. Thereby, we again corrected for the number of tested pairs by calculating FDR values.

The data used was fully anonymized prior to access and analysis. Ahead of data access and analysis, this study was reviewed and approved by the ethics committee of the Leipzig University Medical Faculty (reference number 219/20ek).

## Results

From March 20 to April 17 2020, naso- or oropharyngeal swabs for SARS-CoV-2 RT-PCR were performed in 1460 patients presenting to the Covid-19 outpatient clinic of the Leipzig University Hospital. Of these, 91 (6.2%) patients had a positive RT-PCR and were therefore eligible for inclusion. Two (2.2%) patients were directly admitted to the inpatient department, and a further two (2.2%) patients were lost to follow-up, resulting in a total of 87 patients with confirmed Covid-19 included in the final analysis (**Fig 1**). **Fig 2** shows the age distribution of all patients tested for Covid-19 distinguished by PCR results. Of note, the proportion of test-positive children and adolescents ($\leq 20$ years) was at a comparable level to adults, whilst the proportion of patients diagnosed with Covid-19 above the age of 50 years tended to be elevated. Overall, differences in the percentage of test-positives were not statistically significant in our sample (p = 0.20).

The median age of study participants was 37.0 years (interquartile range (IQR) = 28.5–53), and 48 (55.2%) were female. Patients presented for testing after a mean of 5.9 days (IQR = 2.0–8.5) with a minimum of 0 and a maximum of 24 days following the onset of symptoms. The mean duration between symptom onset and testing decreased from initially 10 days to an average of 2 days during the study timeframe (**S1 Fig**). The majority of patients were classified as early presenters (n = 65, 74.7%) defined as patients presenting for testing during the first 7 days following the onset of symptoms.

Most common symptoms reported by study participants were fatigue (n = 64, 74%), cough (n = 58, 67%), and hyposmia/hypogeusia (n = 44, 51%) (**Table 1**). While the proportion of patients suffering from fatigue and hyposmia/hypogeusia decreased steadily during the course of illness to 26% (n = 21/81) and 15% (n = 12/81), respectively, cough was the most persistent symptom, reported by 32% (n = 26/81) of patients during the third week of illness. Fever, defined as a body temperature $\geq 38.0°C$, was reported at least once by 29% (n = 25/87) of patients. Other less frequent symptoms included rhinitis (30%), headache (30%), dyspnea

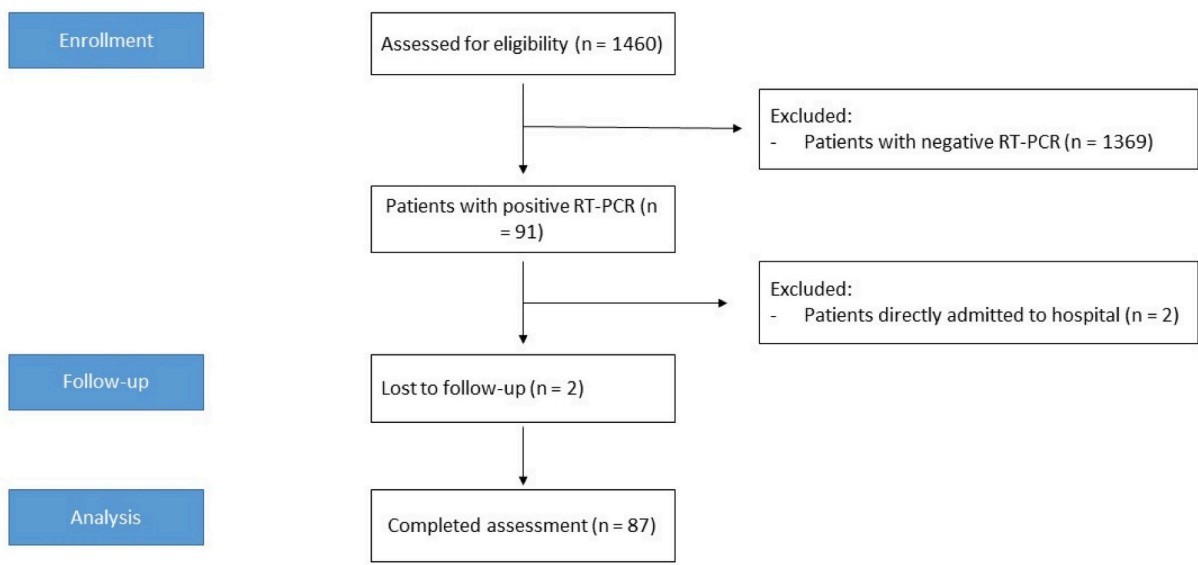

**Fig 1. Diagram of enrollment, follow-up, and analysis.**

(21%), sore throat (20%), diarrhea (9%), and chest pain (7%) (**Table 1**). The distribution and frequency of symptoms in late presenters showed little differences compared with early presenters. However, late presenters reported a significantly longer duration of symptoms. Differences were strongest during the third week for hyposmia/hypogeusia observed in 43% (n = 9/21) of late vs. 5% (n = 3/60) of early presenters (p = 0.0002, FDR = 0.004), and for cough observed in 52% (n = 11/21) of late vs. 25% (n = 15/60) of early presenters (p = 0.03, FDR = 0.24) (**Fig 3**). A detailed overview of type and duration of symptoms is given in **Fig 4** for early presenters and in **Fig 5** for late presenters.

Correlation of symptoms observed in individual patients are shown in **Fig 6**. Fatigue correlated significantly with cough in the same patient at a moderate level (r = 0.33). Weaker, but still statistically significant correlations were found for fatigue and fever (r = 0.2), fatigue and hyposmia/hypogeusia (r = 0.19), fatigue and dyspnea (r = 0.15), fever and headache (r = 0.15), as well as diarrhea and inpatient admission (r = 0.15).

Three (3.5%) patients were asymptomatic during the entire observation period. One patient with previous direct contact to a confirmed Covid-19 case developed symptoms only three days after confirmed diagnosis. A linear decline of symptoms was found in early and late presenters, and the majority of previously symptomatic patients reported to have recovered completely by day 14 (n = 37/70, 52.9%).

Due to disease progression, five (6%) patients (three males, two females) required inpatient care. Two of these patients were admitted to a hospital during the first week of illness, and the other three patients during the second week. Inpatient care lasted for a maximum of five days in four patients, while the fifth patient was admitted for 10 days due to a persistently positive

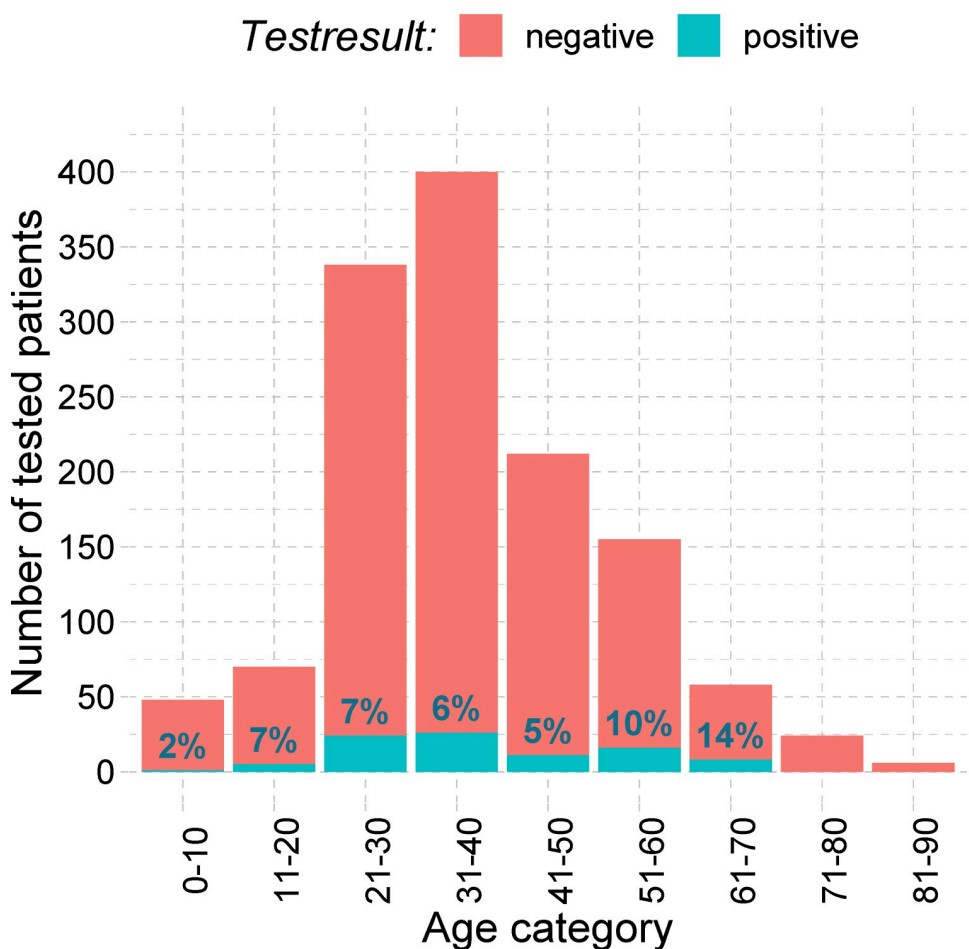

**Fig 2. Number of tested patients and observed test-positives per age group (n = 1460).**

**Table 1. Symptoms reported by outpatients diagnosed with coronavirus disease 2019 (Covid-19).**

| Symptom | All patients | Early presenters | Late presenters | p-value early vs. late |
|---|---|---|---|---|
| fatigue | 64 (74%) | 52 (80%) | 12 (55%) | 0.027[§] |
| cough | 58 (67%) | 45 (69%) | 13 (59%) | 0.44 |
| hyposmia/ hypogeusia | 44 (51%) | 33 (51%) | 11 (50%) | 1.00 |
| rhinitis | 32 (37%) | 24 (37%) | 8 (36%) | 1.00 |
| headache | 26 (30%) | 19 (29%) | 7 (32%) | 0.80 |
| fever ≥ 38˚ Celsius | 25 (29%) | 19 (29%) | 6 (27%) | 1.00 |
| dyspnea | 18 (21%) | 14 (22%) | 4 (18%) | 1.00 |
| sore throat | 17 (20%) | 11 (17%) | 6 (27%) | 0.35 |
| subjective deterioration | 13 (15%) | 10 (15%) | 3 (14%) | 1.00 |
| diarrhea | 8 (9%) | 6 (9%) | 2 (9%) | 1.00 |
| chest pain | 6 (7%) | 4 (6%) | 2 (9%) | 0.64 |
| inpatient admission | 5 (6%) | 4 (6%) | 1 (5%) | 1.00 |

Early presenters–test within 7 days following the onset of symptoms; late presenters–test later than 7 days following the onset of symptoms (§ false discovery rate = 0.32).

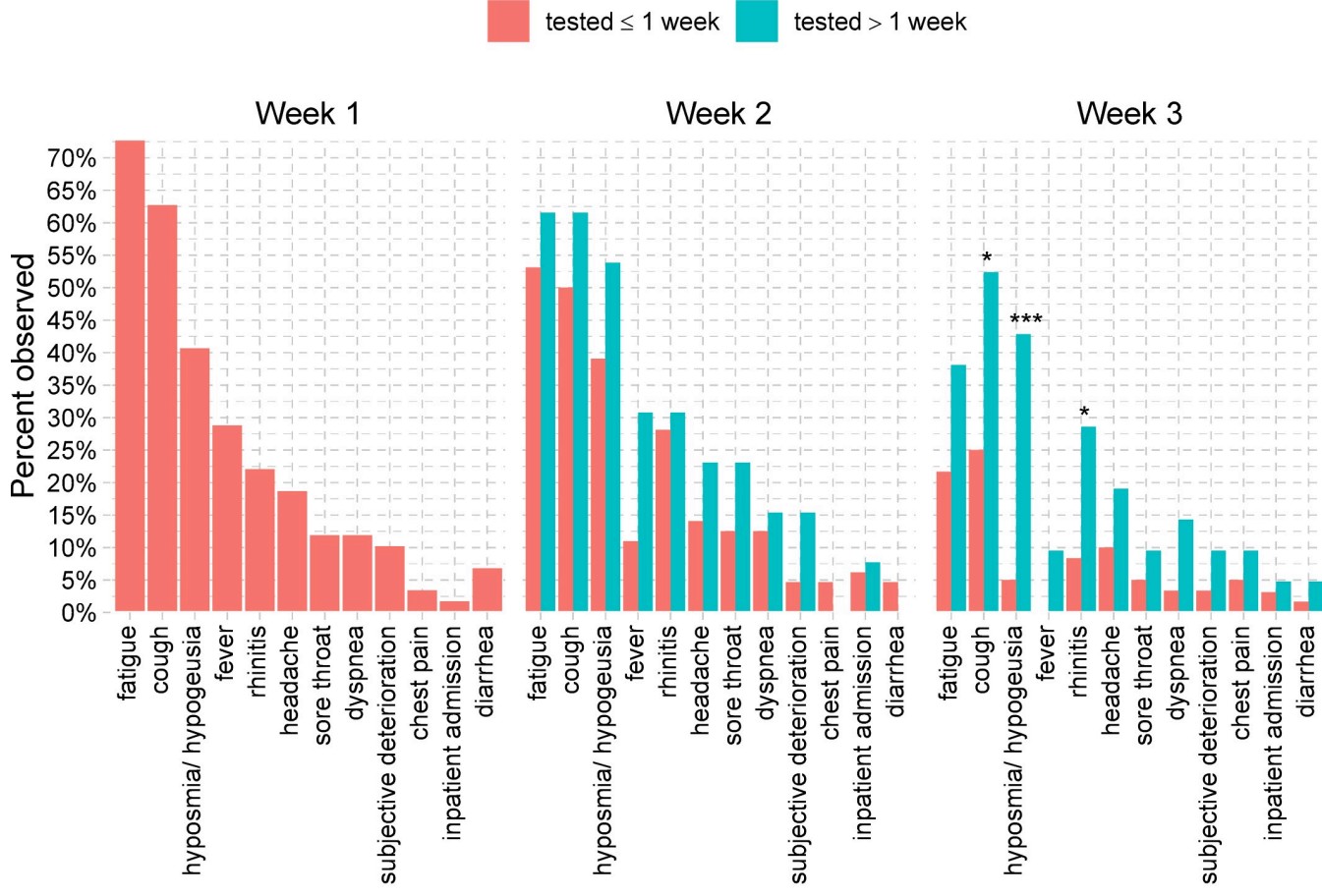

**Fig 3. Symptoms in the first three weeks after symptom onset for patients presenting early ($\leq$ 7 days after symptom onset) or late ($>$ 7 days after symptom onset).** Symptoms typically persisted longer in late presenters compared to early presenters. This was observed strongest for hyposmia/hypogeusia ($p<0.001$, FDR$<0.05$, indicated by ***), cough and rhinitis ($p<0.05$, FDR$<0.3$, indicated by *).

SARS-CoV-2 PCR. Two patients were diagnosed with bilateral pneumonia, of which one experienced intermittent hypoxia. None of the study participants required intensive care treatment.

## Discussion

To the best of our knowledge, this is the first systematic analysis of symptoms reported by outpatients with mild to moderate Covid-19 in Europe. So far, most available reports on the clinical course of Covid-19 refer to severely affected patients requiring in-hospital care. Besides case or cluster reports, to date, only few studies systematically analyzed symptoms occurring in outpatients with Covid-19 [15, 16]. Our study gives important insights into type and duration of self-reported symptoms in 87 outpatients diagnosed with Covid-19. Symptoms were generally mild to moderate, and only a small proportion of 6% required in-hospital treatment.

With a median age of 37 years, our study population was rather young, compared to other studies [17]. This is in line with the officially reported age distribution of Covid-19 cases in Germany at the time when the study was conducted. The two largest age groups of patients infected were reported to be 35–59 and 15–34 year-olds, accounting for 42% and 24% of all

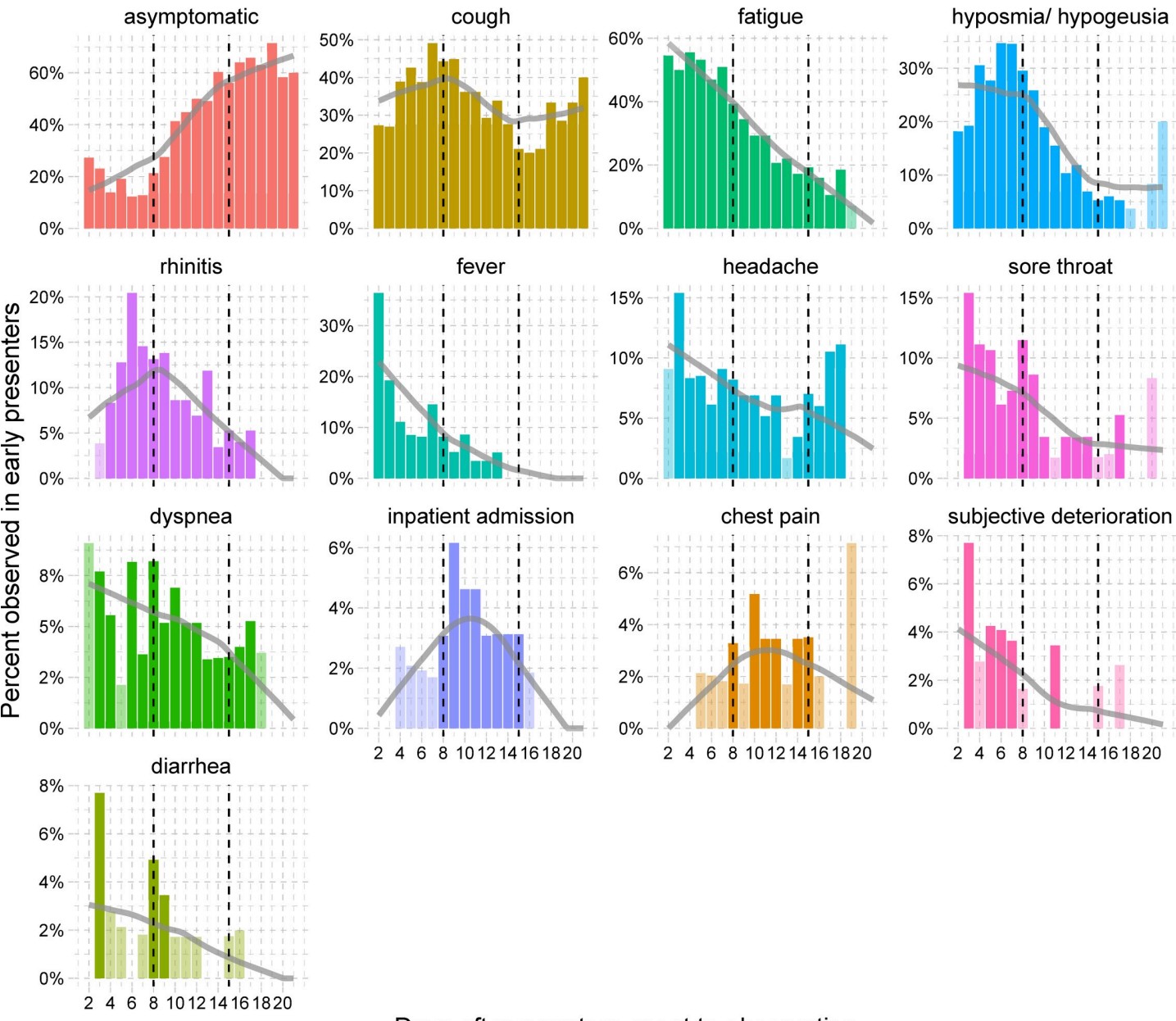

**Fig 4. Time-course of symptoms observed for early presenters patients (≤ 7 days from symptom onset to testing) (n = 65).** Observations are arranged by frequency. Dashed lines indicate the start of the second and third week after symptom onset. Bars in lighter color represent observations, where only a single patient developed a specific symptom. Grey trend lines are trends calculated as loess-smoother.

test-positives, respectively. We attribute this mainly to the mode of infection on the one hand, as a large proportion of patients acquired the infection during skiing holidays in Austria or South Tyrolea, Italy, which are popular destinations among young adults and well known for so-called *après-ski* get-togethers. On the other hand, evidence suggests that the older generation, which is prone to severe courses of Covid-19 disease, was highly compliant to so-called lock-down measures in the greater Leipzig region. Of note, our analysis does not include residents of nursing or retirement homes.

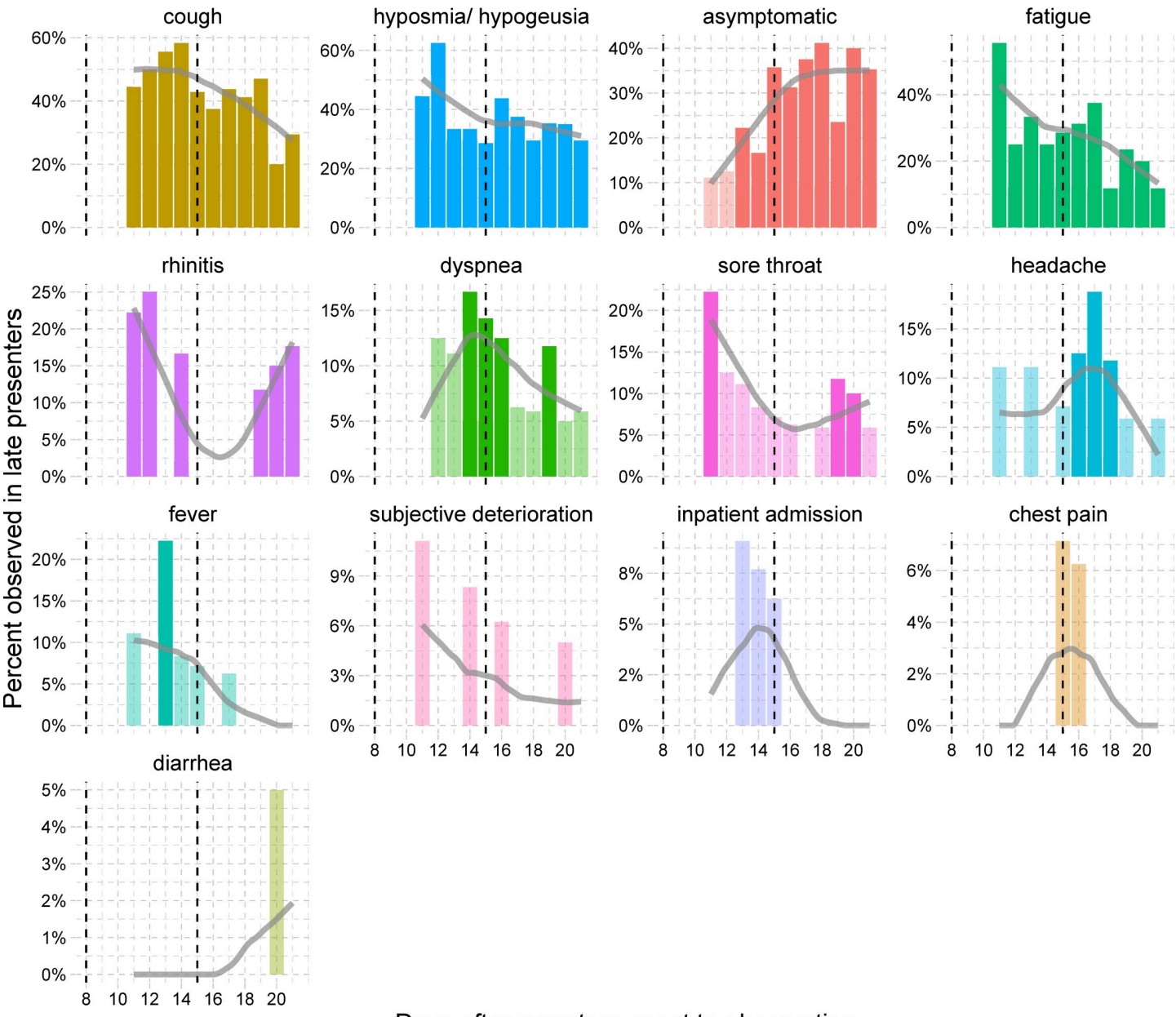

**Fig 5. Time-course of symptoms observed for late presenters (> 7 days from symptom onset to testing) (n = 22).** Observations are arranged by frequency. Dashed lines indicate the start of the second and third week after symptom onset. Bars in lighter color represent observations, where only a single patient developed a specific symptom. Grey trend lines are trends calculated as loess-smoother.

During the beginning of our study period, when the Covid-19 pandemic had just started to affect our region as a whole and media coverage began to urge people to take symptoms seriously, patients presented for testing after a mean of 10 days after symptom onset. One patient —in retrospect most likely one of the first people infected with SARS-CoV-2 in Leipzig—presented for testing 24 days after the onset of symptoms. Most likely, many patients with mild symptoms did not see the necessity for testing at the time. In line with our observation, a comparable study from South Korea reported that outpatients with mild to moderate symptoms

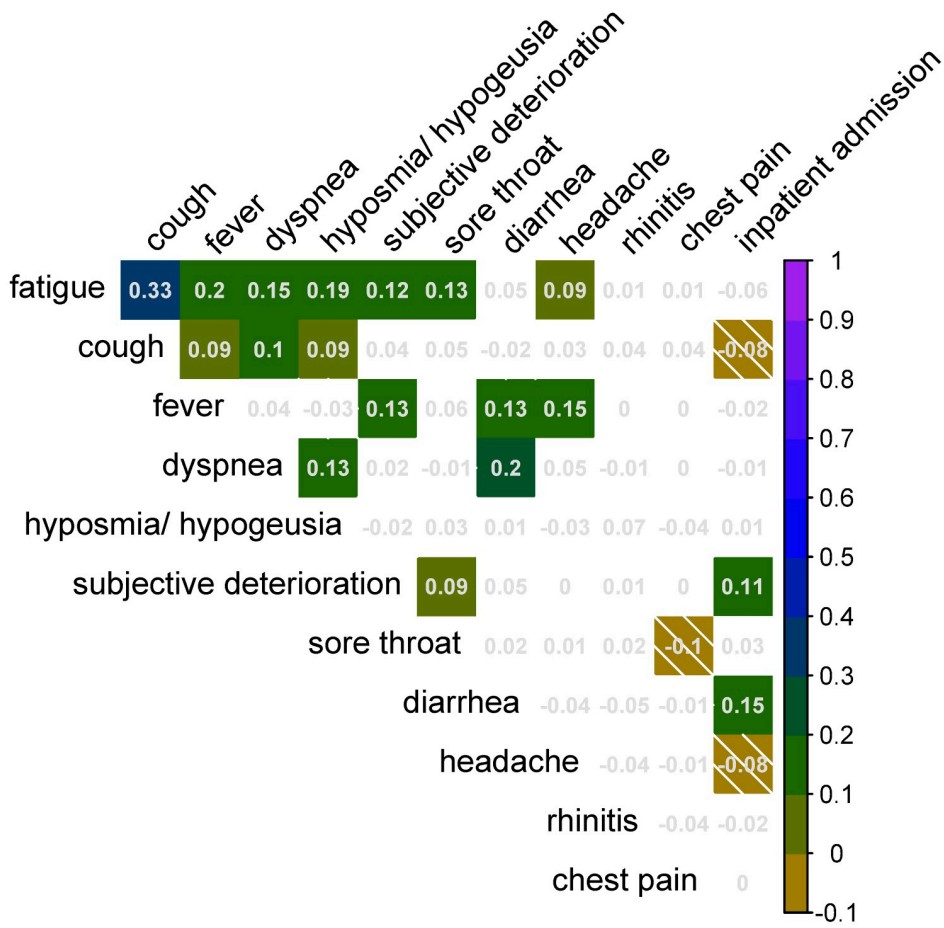

**Fig 6. Positive and negative correlation of symptoms observed in individual patients.** Shown are all pairwise correlations, significant correlations are indicated by a background color (FDR≤0.05). Color codes in the color bar represent the size of the correlation coefficient r. Anti-correlation of symptoms (r≤0) is emphasized with diagonal stripes.

waited for a mean of 14 days bevor undergoing testing [15]. It is important to be aware of this, as these patients are infectious to others and delayed diagnosis results in delayed public health measures such as contact tracing. As awareness grew, the time between symptom onset and testing shortened significantly in our study population, demonstrating the principal efficacy of public information policies. Public awareness and education should include emphasizing the

necessity to undergo testing independently of symptom severity in order to break the chain of transmission as early as possible. Our results contribute to strengthening these recommendations by providing information specifically on outpatients with mild symptoms.

Detecting and isolating patients with Covid-19 early as well as tracing potential contacts as soon as possible are important measures for a successful containment strategy. Thus, sensitive and easily accessible screening mechanisms and reliable indicators of SARS-CoV-2 infection are essential. To date, body temperature measurement is a widely used screening method for Covid-19 in highly frequented public places such as airports, hotels, convention centers or hospitals, as previous reports found fever to be one of the most common symptoms (> 70%) [17, 18]. However, these reports predominantly refer to hospitalized patients. In contrast, within our study population less than a third of patients reported fever. A similar Korean study assessing patients with mild to moderate courses of Covid-19 found even lower proportions (< 20%) [15]. Fever seems to be common in severe and critical courses of the disease, while only a minority of mildly to moderately affected patients report elevated body temperature. Thus, assessing the presence of fever only may be insufficient for Covid-19 screening in public places.

In our study, cough was one of the most frequent symptoms, reported by two thirds of participants. Fatigue was the only symptom reported by even more patients. Loss of smell or taste, reported by variable proportions of patients from 5% to 88% in previous studies, was the third most common symptom in our study population, occurring in half of the patients [7–10]. Similar to a study by Bocksberger et al., there was no significant correlation between hyposmia/hypogeusia and the occurrence of rhinitis [8]. Since fatigue is a rather unspecific symptom, a brief query on the presence or absence of cough and/or hyposmia/hypogeusia additionally to the measurement of body temperature may be a more targeted way to detect mild cases of Covid-19 than measuring body temperature alone. In further support of this, hyposmia/hypogeusia persisted much longer than fever and was quite frequent in the third week of illness in late presenters, whereas the presence of fever quickly declined during the first week.

Concerning the spread of SARS-CoV-2, a further relevant factor is transmission by pre- or asymptomatic cases, which is reported from clusters worldwide with incidence rates of 2% to 52% [19]. Most recently, the full magnitude of SARS-CoV-2 transmission by pre- and asymptomatic Covid-19 cases were reported from the outbreak on the cruise ship "*Diamond Princess*" in Japan [20]. Almost all of our patients reported symptoms at any point in time during the course of disease. However, the low number of asymptomatic infections in our study population is explained by the testing policy, which only sought to test asymptomatic persons if they worked in system-relevant areas and had direct contact (> 15 min face-to-face or physical contact such as handshakes) to confirmed Covid-19 cases.

Main limitations of our study include the fact that our analysis of symptoms is based on self-reporting only, and that underlying medical conditions were not systematically assessed. Furthermore, while our study contributes to filling the gap in research on Covid-19 in young out-patients, larger cohort studies will be necessary to translate these findings into effective clinical screening mechanisms for Covid-19.

To conclude, our rather young study population was only mildly to moderately affected by Covid-19, and fever occurred less frequently than previously assumed. This suggests that elevated body temperature alone may be an insufficient indicator of SARS-CoV-2 infection.

## Supporting information

**S1 Fig. Days from symptom onset to testing by polymerase chain reaction (n = 87).**
(TIF)

**S1 File.**
(PDF)

**S2 File.**
(PDF)

## Acknowledgments

We would like to acknowledge all study participants. We especially thank the team of nurses, medical students and doctors at our Covid-19 clinic for their dedicated patient care and tireless support during these challenging times.

## Author Contributions

**Conceptualization:** Anne Schneider, Amrei von Braun.

**Data curation:** Anne Schneider, Franziska Lordick.

**Formal analysis:** Anne Schneider, Holger Kirsten.

**Methodology:** Holger Kirsten.

**Project administration:** Anne Schneider.

**Supervision:** Amrei von Braun.

**Validation:** Amrei von Braun.

**Visualization:** Holger Kirsten.

**Writing – original draft:** Anne Schneider.

**Writing – review & editing:** Holger Kirsten, Florian Lordick, Christoph Lübbert, Amrei von Braun.

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
