## [Decision Letter · Decision Letter 0]

8 Jan 2021

PONE-D-20-20307

Covid-19 in outpatients - is fever a useful indicator for SARS-CoV-2 infection?

PLOS ONE

Dear Dr. von Braun,

Thank you for submitting your manuscript to PLOS ONE. After careful consideration, we feel that it has merit but does not fully meet PLOS ONE’s publication criteria as it currently stands. Therefore, we invite you to submit a revised version of the manuscript that addresses the points raised during the review process.

The topic is quite interesting. However, the main limitation of the present investigation is represented by the small number of patients. we suggest to add some more patients to reach a stronger statistolical power. 

We look forward to receiving your revised manuscript.

Kind regards,

Chiara Lazzeri

Academic Editor

PLOS ONE

Journal Requirements:

Reviewers' comments:

Reviewer's Responses to Questions

**Comments to the Author**

1. Is the manuscript technically sound, and do the data support the conclusions?

Reviewer #1: Partly

2. Has the statistical analysis been performed appropriately and rigorously? 

Reviewer #1: Yes

3. Have the authors made all data underlying the findings in their manuscript fully available?

Reviewer #1: Yes

4. Is the manuscript presented in an intelligible fashion and written in standard English?

Reviewer #1: Yes

5. Review Comments to the Author

Reviewer #1: The authors systematically analyzed the symptoms from outpatients with mild to moderate COVID-19 and found that fever was not a common symptom in mild cases. This study is interesting. However, the following major concerns need to be sorted out before it is accepted.

1.The sample size of confirmed covid-19 was small (n=91). Could you add more confirmed mild cases for statistical analysis?

2.The descriptive bias existed as all these data were obtained by telephone interview. Again, it would be better if the sample size is increased.

6. PLOS authors have the option to publish the peer review history of their article (what does this mean?). If published, this will include your full peer review and any attached files.

Reviewer #1: No

---

## [Author Response · Author response to Decision Letter 0]

13 Jan 2021

We would like to express our sincere gratitude to the reviewer for their precise review of our manuscript. The comments provided to us are highly appreciated. Kindly find our reply below.

Reviewer #1: The authors systematically analyzed the symptoms from outpatients with mild to moderate COVID-19 and found that fever was not a common symptom in mild cases. This study is interesting. However, the following major concerns need to be sorted out before it is accepted.

1.The sample size of confirmed covid-19 was small (n=91). Could you add more confirmed mild cases for statistical analysis?

Response: Thank you for your kind introductory sentence. We completely agree that our sample size is not large enough to make binding statements on clinical screening mechanisms for Covid-19. Larger cohort studies will be necessary to identify such mechanisms based on symptom evaluations in all age groups. However, our study does contribute valuable clinical data on mild to moderate cases which continues to be rare compared to data on severely effected patients. 

In full agreement with the reviewer, we have added this point to the section on study limitations (page 12, lines 262-266): “(…) while our study contributes to filling the gap in research on Covid-19 in young out-patients, larger cohort studies will be necessary to translate these findings into effective clinical screening mechanisms for Covid-19.”

We will unfortunately not be able to include more patients into the analysis as our hospital shifted to in-patient care on the wards and the intensive care units only. Unlike during the first months of the pandemic, out-patient care is now fully covered by general practitioners.

2.The descriptive bias existed as all these data were obtained by telephone interview. Again, it would be better if the sample size is increased.

Response: Thank you for pointing this out. Yes, follow-up data was obtained by telephone interview as these patients were subject to official quarantine measures and clinical consultations at our hospital were organized only if necessary. The descriptive bias is reflected in the study limitations (page 12, line 262-263): “Main limitations of our study include the fact that our analysis of symptoms is based on self-reporting only, and that underlying medical conditions were not systematically assessed. “

On behalf of all co-authors, I thank you for considering our work in its revised version.

---

## [Editor Report · Decision Letter 1]

18 Jan 2021

Covid-19 in outpatients - is fever a useful indicator for SARS-CoV-2 infection?

PONE-D-20-20307R1

Dear Dr. von Braun,

We’re pleased to inform you that your manuscript has been judged scientifically suitable for publication and will be formally accepted for publication once it meets all outstanding technical requirements.

Kind regards,

Chiara Lazzeri

Academic Editor

PLOS ONE
---

## [Editor Report · Acceptance letter]

26 Jan 2021

PONE-D-20-20307R1 

Covid-19 in outpatients – is fever a useful indicator for SARS-CoV-2 infection? 

Dear Dr. von Braun:

I'm pleased to inform you that your manuscript has been deemed suitable for publication in PLOS ONE. Congratulations! Your manuscript is now with our production department. 

Kind regards, 

on behalf of

Dr. Chiara Lazzeri 

Academic Editor

PLOS ONE